# An Overview of PARP Resistance in Ovarian Cancer from a Molecular and Clinical Perspective

**DOI:** 10.3390/ijms241511890

**Published:** 2023-07-25

**Authors:** Nicoletta Cordani, Tommaso Bianchi, Luca Carlofrancesco Ammoni, Diego Luigi Cortinovis, Marina Elena Cazzaniga, Andrea Alberto Lissoni, Fabio Landoni, Stefania Canova

**Affiliations:** 1School of Medicine and Surgery, University of Milano-Bicocca, 20900 Monza, Italy; t.bianchi4@campus.unimib.it (T.B.); lucaammoni@gmail.com (L.C.A.); marina.cazzaniga@unimib.it (M.E.C.); andreaalberto.lissoni@unimib.it (A.A.L.); fabio.landoni@unimib.it (F.L.); 2Clinic of Obstetrics and Gynecology, Fondazione IRCCS San Gerardo dei Tintori, University of Milano-Bicocca, 20900 Monza, Italy; 3Medical Oncology Unit, Fondazione IRCCS San Gerardo dei Tintori, 20900 Monza, Italy; diegoluigi.cortinovis@irccs-sangerardo.it; 4Phase 1 Research Centre, Fondazione IRCCS San Gerardo dei Tintori, 20900 Monza, Italy

**Keywords:** epithelial ovarian cancer, PARP inhibitors, resistance

## Abstract

Epithelial ovarian cancer (EOC), a primarily high-grade serous carcinoma (HGSOC), is one of the major causes of high death-to-incidence ratios of all gynecological cancers. Cytoreductive surgery and platinum-based chemotherapy represent the main treatments for this aggressive disease. Molecular characterization of HGSOC has revealed that up to 50% of cases have a deficiency in the homologous recombination repair (HRR) system, which makes these tumors sensitive to poly ADP-ribose inhibitors (PARP-is). However, drug resistance often occurs and overcoming it represents a big challenge. A number of strategies are under investigation, with the most promising being combinations of PARP-is with antiangiogenetic agents and immune checkpoint inhibitors. Moreover, new drugs targeting different pathways, including the ATR-CHK1-WEE1, the PI3K-AKT and the RAS/RAF/MEK, are under development both in phase I and II–III clinical trials. Nevertheless, there is still a long way to go, and the next few years promise to be exciting.

## 1. Introduction

Epithelial ovarian cancer (EOC), a primarily high-grade serous carcinoma (HGSOC), is one of the major causes of high death-to-incidence ratios of all gynecologic cancers. Cytoreductive surgery and platinum-based chemotherapy represent the principal treatment for EOC [1]. Gynecologic cancers were initially treated with platinum chemotherapy, leading to an accumulation of deoxyribonucleic acid (DNA) double-strand breaks with the activation of cellular DNA damage response (DDR) pathways [2]. Poly ADP-ribose inhibitors (PARP-is) were initially approved by the Food and Drug Administration (FDA) only for EOC with BRCA-1 and BRCA-2 mutations. Regardless of BRCA mutation status, patients affected by relapsed EOC showing sensitivity to platinum-based therapy are treated with PARP-is following National Comprehensive Cancer Network guidelines (https://www.nccn.org/professionals/physician_gls/pdf/ovarian.pdf (version 5 2022, accessed on 16 September 2022). However, drug resistance often occurs, and overcoming it represents a big challenge. Furthermore, given their target-based mechanism of action, PARP-is provide the best therapeutic advantage in homologous recombination repair (HRR)-defective (HRD) tumors. Therefore, identification of HRR impairment with a high-fidelity test would identify those patients who could benefit most from PARP-is and avoid over-treatment and drug-related side effects [3,4].

## 2. PARP

Poly(ADP-ribose) polymerases (PARPs) or diphtheria toxin-type ADP-ribose transferases (ARTDs) constitute a protein family: 16 for mice and 17 for humans [5]; they are localized in the nucleus and in the cytosol. PARPs are multidomain enzymes sharing the same catalytic domain that shows structural homology to other ADP-ribosyl transferase enzymes. PARP1 and 2 are divided into two principal domains: the DNA-binding and the catalytic domains. PARP1 comprises six domains [6]. The N-terminal domain of PARP1, a metalloenzyme with the capacity to specifically link Zn molecules [7], is responsible for binding damaged DNA with Zn fingers (Zn1, Zn2, Zn3) in cooperation with the Trp-Gly-Arg (WGR) domain (Figure 1). Instead, PARP1 Zn fingers and the BRCT domain (breast cancer-susceptibility protein C-terminal domain) contain auto-modification sites. In the C-terminus, there are the WGR domain and the catalytic domain containing the helical domain (HD) and an ART domain [8].

PARP enzymes are the effectors of ADPRylation, a dynamic and reversible post-transcriptional modification of proteins where ADP-ribose units are transferred from nicotinamide adenine dinucleotide (NAD+) to proteins and nucleic acids [9,10]. PARP1 and PARP-2 use β-NAD+ as a substrate, and synthesize and transfer ADP-ribose polymers onto glutamate, aspartate, or lysine residues of acceptor proteins, building poly (ADP-ribose) (PAR) polymers and modifying their functional properties [11]. On the other hand, PARP3 catalyzes mono (ADP)-ribosylation [12]. PARPs have many domains that can interact with DNA protein repair or other proteins such as transcription factors. Other domains can attach directly to damaged nucleic acids, as reported in 2013 by Farrè et al. [9,13]. Previous studies support the view that PARP1 and PARP2 are crucial in maintaining genome integrity, because in mice with deletions in either PARP1 or PARP2, defects in the maintenance of chromosome structure and DNA repair have been shown [5].

### PARP Functions

When a cell undergoes metabolic changes under the influence of both exogenous and endogenous (physical and chemical) factors, damage can occur and it must be repaired. These processes are active at various times throughout the cell cycle, including replication. PARP1, 2, and 3 principally act in the nucleus and are activated and involved after DNA damage response (DDR), playing a key role in DNA repair [14].

In this pathway, the following processes are included:-Nucleotide excision repair (NER), a highly conserved DDR process that corrects a variety of genomic lesions [15];-Base excision repair (BER), the process in which DNA damages are removed by PARP1, 2, and 3 interaction with the damaged DNA in the nucleosomal context [16,17];-Mismatch repair (MMR) pathway that leads to the removal of DNA polymerase errors that appear during replication [18];-Homologous recombination (HR) that controls S and G2 phases [19]. The end effector in HR, RAD51 recombinase, is recruited by BRCA1 and BRCA2.-Non-homologous end joining recombination (NHEJ) is a different route for repairing DNA damage during the G1 phase of the cell cycle [19].

The DNA damage repairing mechanism is divided into three general phases: (1) detection of DNA damage, (2) recruitment of co-factors, and (3) regulation of biochemical activities (Figure 2).

DDR proteins are recruited by PARP’s modification by auto-PAR [13,20] to initiate single-stranded break (SSB) and double-stranded break (DSB) repair [21]. BRCA1 and BRCA2, known as breast cancer susceptibility genes, are important mediators in HR, which is one of the most relevant processes for repairing DDS [22]. The wild-type BRCA gene is a crucial effector in HR with the aid of PARP1/2 and it is present in healthy cells in at least one copy. Since blocked replication forks cannot be restarted without the help of PARP1/2, SSB inevitably lead to more severe DSB; hence, tumor cells lacking active BRCA1 or BRCA2 are sensitive to a PARP inhibitor molecule [23]. Nowadays it is well known that PARP-is induce apoptosis in BRCA-deficient cancer cells, so this mechanism is evidence that these molecules are relevant as targeted therapy in BRCA-mutated cancer [24].

## 3. HRD and Ovarian Cancer

Thanks to the TCGA project, which performed a comprehensive characterization of HGSOC molecular features through genomic, epigenomic, and transcriptomic analysis of 489 high-grade serous ovarian adenocarcinomas, HRR deficiency was found as a hallmark and a crucial therapeutic target of HGSOC [25]. BRCA1 and BRCA2 are the most frequently mutated HRR genes in HGSOC, both at the germline and somatic levels. Germline mutations of BRCA-1 and BRCA-2 (gBRCA1-2 mutations) are responsible for hereditary breast and ovarian cancer (HBOC) syndrome and give a cumulative lifetime risk of developing HGSOC of 44% and 17% respectively [25,26]. According to TGCA, gBRCA1-2 mutations occur in 9% and 8% of cases, respectively [25]. Somatic mutations of BRCA1-2 (sBRCA1-2 mutations) are present in a further 5–7% of cases; they seem to be an early clonal event in tumorigenesis and thus are presented in all tumor cells, as well as TP53 mutations [25,27,28]. Comprehensively, gBRCA1-2 and sBRCA1-2 mutations account for about 20% of HGSOC. As previously debated, the HRR system consists of a complex pathway in which many proteins with different roles are involved. In HGSOC, HRD is displayed also in BRCA1-2 wild-type (wtBRCA1-2) tumors because of biallelic mutations of other HRR genes as well as their epigenomic silencing. HRD in wtBRCA1-2 tumors mostly concerns both germline and somatic mutations in PTEN, RAD51C, RAD51D, BRIP1, PALB2, and CHEK1-2, which account for about 15–20% of HGSOC [29], and epigenomic silencing of HRR through BRCA1-2 and RAD51C promoter hypermethylation, which account, according to TCGA data and more recent translational studies, for 11–19%, 5%, and 2% of HGSOC, respectively [25,30,31]. sBRCA, gBRCA, and HRD wtBRCA1-2 tumors due to genomic HRR gene mutations share a common clinical behaviour named “BRCA-ness” or HRD phenotype, characterized by an improved survival and a better response to platinum-based therapy, compared to their HRR-proficient (HRP) counterparts [29,32,33]. Besides giving a better prognosis and a better response to platinum-based chemotherapy, HRD is a predictive biomarker of response to PARP-is. PARP-is exploit their antitumoral activity, blocking PARP-mediated SSB repair, causing accumulation of DNA SSBs, replication fork collapse, and accumulation of DSBs. In normal cells, DSBs are efficiently repaired by HRR, but this does not occur in HRD tumor cells, which rely on error-prone DSB repair systems such as NHEJ, leading to accumulation of DNA lesions and finally resulting in cell death [34,35]. PARP-is were initially utilized as maintenance therapy for HGSOC treatment in the relapse setting. The clinical efficacy of PARP-is in the relapse setting raised the question of whether they could be used as maintenance treatment after or alongside first-line chemotherapy. Three clinically authorized PARP-is are currently available for HGSOC treatment in both first-line and relapse settings: olaparib, niraparib, and rucaparib. A further molecule, veliparib, is under authorization process by the FDA and European Medicines Agency (EMA). All the way up talazoparib, a PARPi approved for breast cancer treatment, all these molecules act through a common mechanism of action known as “synthetic lethality”, a term indicating a process by which inactivation of two molecular pathways leads to cell death, where single inactivation of either pathway is not lethal.

### 3.1. PARP-Is Resistance

In recent years, the use of PARP-is has progressively increased in routine clinical practice both as a first- and second-line maintenance therapy. Although they usually elicit a good initial response, many patients develop disease progression or relapse as tumor cells become resistant to PARP-is. PARP-is resistance can occur via three general mechanisms: restoration of HRR, mitigation of replication stress, and upregulation of drug efflux pumps.

### 3.2. Restoration of HRR

Restoration of the HRR system is the most common acquired mechanism of resistance to PARP-is and can occur through various processes. Direct restoration of HRR is mediated by reversion mutations that restore the wild-type function of mutated genes as promoters of demethylation of HRR genes. Reversion mutations are somatic substitutions of base replacements, or insertions/deletions close to the main mutation of the protein, that restore the open reading frame (ORF) of the gene and functional protein. They represent a key mechanism of PARP-is resistance, shifting the neoplastic cell from HRR deficiency to efficiency. Reversion-to-wild-type mutations have been first described in BRCA1-2 genes. Restoration of BRCA1-2 function is mediated by genetic events that compensate the frameshift, occurring in case of protein-truncating mutation, and can lead to transcription of a functional or near-to-functional BRCA1-2 protein. BRCA1-2 reversion mutations were originally assessed in vitro. For instance, Edward et al. [36] described the restoration of HRR system function in a BRCA2-mutated pancreatic cell line as a result of an acquired mutation restoring the open reading frame. Similarly, Sakai et al. [37] reported acquired resistance to cisplatin and PARP-is in BRCA2 breast and pancreatic cancer cell lines through acquisition of secondary genetic events restoring wild-type BRCA2 function. Barber et al. [38] performed an NGS-based assay on FFPE tissue samples of treatment-naïve and -resistant tumor material from patients with germline BRCA2 mutations affected by ovarian cancer with olaparib-acquired resistance. The analysis evidenced tumor-specific BRCA2 secondary mutations in olaparib-resistant metastasis, restoring full-length BRCA2 functional protein. Furthermore, Lin et al. [39] analyzed circulating cell-free DNA (ccfDNA) derived from plasma samples collected before rucaparib treatment in 112 patients with gBRCA1-2 or sBRCA1-2-mutated high-grade ovarian carcinoma enrolled in ARIEL2. BRCA reversion mutations were identified in cfDNA from 18% of platinum-refractory and 13% of platinum-resistant cancers, but only in 2% of platinum-sensitive cancers. Patients without BRCA reversion mutations detected in pre-treatment cfDNA had significantly longer rucaparib progression-free survival than those with reversion mutations (9.0 vs. 1.8 months). For instance, the authors found four different BRCA1 reversion mutations through cfDNA sequencing in one patient with platinum-resistant cancer and a sBRCA1 mutation. These reversion mutations they found all involved a single amino acid change and deletions restoring the ORF (Table 1) [39].

A metanalysis by Tobalina et al. showed that reversion mutations seem to be more frequent in BRCA2 than BRCA1 and that deletions account for the majority of secondary mutations both in BRCA1 and BRCA2, as a consequence of the use of error-prone DNA repair mechanisms of end joining, suggesting that NHEJ repair mechanisms could be a relevant player in generating reversions during PARP-is treatment [40]. Importantly, reversion mutations may occur not only in BRCA1-2 but even in other HRR genes. For instance, sequencing of HRR genes in tumor biopsy samples collected before and after rucaparib treatment in 12 patients enrolled in ARIEL2 was performed by Kondrashova et al. [41]. They showed that reversion mutations restored the open reading frame in 5 of 6 biopsies with a pre-treatment truncating BRCA1, RAD51C, or RAD51D mutation. Finally, an additional mechanism of direct restoration of HRR is BRCA1 promoter methylation loss. Preclinical studies show that expression of functional BRCA1 protein in patient-derived xenograft (PDX) models of BRCA1-methylated triple-negative breast cancer can be mediated by BRCA1 hypermethylation loss [42]. Additionally, heterozygous methylation of BRCA1 in BRCA1 methylated ovarian cancer PDX models is associated with PARP-is resistance, whereas complete BRCA1 methylation predicts clinical response to PARP-is [43]. According to these data, Swisher et al. exploited a post hoc exploratory biomarker analysis on a large dataset of archival tissue samples from patients enrolled in the ARIEL 2 study to identify molecular mechanisms resulting in rucaparib sensitivity and resistance. Firstly, they demonstrated that high BRCA1 methylation before initiating rucaparib was associated with a better progression-free survival (PFS), with a median PFS comparable to that of BRCA-mutated HGOC patients without reversion mutations, whereas hypomethylated patients had a median PFS similar to that of patients with BRCA reversion mutations. Secondly, analysis of chemo-naïve and pre-rucaparib treatment tumor biopsies showed that a decrease in methylation correlates with poor response to rucaparib; the objective response rate (ORR) among patients who maintained high methylation was 38%, whereas no responses were observed among patients with methylation decrease or loss, indicating that methylation plasticity is a key mechanism of resistance to PARP-is [44]. Indirect restoration of HRR is mediated by oncogenic signalling pathways that drive reset of the HRR pathway regardless of BRCA1-2 activity; particularly, resistance to PARP-is emerges via loss of DNA end-protection in BRCA-1-deficient cells. The first described indirect mechanism of HRR restoration is the loss of 53BP1 activity. Specifically, in vivo studies show that 53BP1 knockout rescues embryonic lethality in mice with BRCA1 deficiency, counteracting the effects of BRCA1 loss on the HRR system and on genomic instability. Furthermore, in vitro studies show that 53BP1 is required for cell death induced by BRCA1 deficiency, since BRCA1-deficient 53BP1-deleted murine embryonic fibroblasts display reduced apoptosis and cell death. HRR initiation involves the resection of DSB extremities in a 5′-3′ direction with the creation of a 3′-tailed end that can be used to invade the homologous sequence on the sister chromatid; 53BP1 binding to chromatid breaks in BRCA1-deficient cells blocks ATM-dependent DNA resection at the break site, leading to DNA repair by NHEJ. Of note, 53BP1 loss restores HRR system function only in BRCA1-deficient cells, whereas it is inactive in case of BRCA2 deficiency [45,46,47,48]. Several other proteins have been identified to be involved in DNA end resection inhibition and whose loss can restore HRR function and confer resistance to PARP-is in BRCA1-deficient cells. Particularly, RIF1, REV7, the shield-in complex and the CST (CTC1-STN1-TEN1) complex act downstream of 53BP1, and both in vivo and clinical studies show that their inactivation through acquired genetic or epigenetic aberrations mediates resistance to PARP-is [49]. Similarly, DYNLL1 acts as an inhibitor of DNA end resection by promoting oligomerization and correct activation of 53BP1 at DNA DSB sites; its inhibition or depletion restores DNA end resection and HRR system in BRCA1-deficient cells, thereby inducing resistance to PARP-is [50,51]. Interestingly, a recent paper has shown that DNA ligase III (LIG3) significantly reduces PARP-is sensibility in BRCA1-mutated cells with depletion of 53BP1. Its overexpression was observed by an immunohistochemistry analysis in selective expansion of LIG3-overexpressing clones exhibiting PARP-is resistance during maintenance treatment. Additionally, and more importantly, loss of LIG3 restores sensitivity to PARP-is both in vitro and in vivo, suggesting that LIG3 could be targeted to bypass resistance to PARP-is [52].

### 3.3. Mitigation of Replication Stress

Both BRCA1 and BRCA2 are not only involved in DNA damage response via the HRR system, but also in replication fork stability during the S phase of the cell cycle. Their absence is associated with the accumulation of stalled replication forks, leading to replication stress and consequently to genomic instability and the hallmark of BRCA-mutated cells, i.e., chemosensitivity [53]. In particular, MRE11 is a nuclease deputed to processing stalled replications forks; its activity, if uncontrolled in case of BRCA1-2 absence, leads to uncontrolled resection of stalled forks and genomic instability [54]. Conversely, restoration of fork stability has been observed to be implicated in the development of PARP-is resistance. Notably, reversion of HRR system impairment and consequent development of resistance to PARP-is through inhibition of the 53BP1 pathway is restricted to BRCA1-deficient tumors, while acquired resistance to PARP-is can develop both in BRCA1- and BRCA2-mutated patients via mitigation of replication stress. As an example, loss of the MLL3/4 complex protein, PTIP, protects BRCA1/2-deficient cells from DNA damage by inhibiting MRE11 recruitment to stalled replication forks, protects DNA from extensive degradation, and is associated with the acquisition of PARP-is and platinum salts resistance independently from HRR restoration [55]. Similarly, the SMARCAL1 complex and RADX are involved in DNA replication stress in BRCA1-2-mutated cells by regulating MRE11 and RAD51 activity, respectively, and their depletion is associated with reduced sensitivity to PARP-is [56,57].

### 3.4. Drug Efflux Pumps

The ABCB1 protein, also known as MDR-1 (multidrug resistance) protein or P-glycoprotein (P-GP), is an ATP-dependent membrane transporter responsible for multiple chemotherapy agents’ efflux and prevention of their accumulation within the tumor cell. Thus, it is a well-known source of chemotherapy resistance in various mammary tumors, and overexpression of P-GP was one the first proposed mechanisms of resistance to PARP-is, as well as to taxanes and platinum salts [58]. Studies on cell lines and animal models have demonstrated that PARP-is are substrates of this transporter. Particularly, Lawlor and colleagues [59] showed resistance to olaparib in drug-resistant cell lines exhibiting upregulation of P-GP. Likewise, Rottenberg et al. evaluated long-term exposure to the PARP-i AZD2281 in BRCA-1-deficient mice, and they evidenced that long-term treatment did result in the development of drug resistance through upregulation of Abcb1a/b genes encoding P-glycoprotein efflux pumps [60]. It is noteworthy in both Lawlor and Rottenmberg studies that reversion of the resistance to PARP-is was obtained when an inhibitor of the efflux pump was administered in combination with the PARP-i, opening the way to the possibility of combining PARP-is with P-GP inhibitors, especially in the recurrent chemo-resistant setting; however, no clinical trials exploring this combination have been designed yet. A further study strengthens the importance of testing this combination: Vaidyanathan et al. [61] tested ovarian cancer cell lines resistant to paclitaxel and olaparib for cross-resistance to other anti-cancer agents and for circumvention of resistance through inhibition of P-GP. Paclitaxel-resistant cells were shown to be cross-resistant to olaparib, doxorubicin, and rucaparib, and resistance could be reversed with the use of verapamil and elacridar, two inhibitors of P-GP. Interestingly, in line with Lawlor’s data, cross-resistance did not affect veliparib, suggesting that this PARP-i is a poor substrate of P-GP, and its use should be explored in chemo-resistant HGSOC with proven P-GP upregulation. One of the proposed mechanisms to explain P-GP overexpression is gene promoter activity enhancement. In a whole genome sequencing analysis of 114 recurrent chemotherapy-resistant HGSOC samples performed by Patch et al. [62], in up to 8% of recurrent HGSOC P-GP promoter fusions and translocations were found, leading to its upregulation. These findings were confirmed by a subsequent paper by Christie and colleagues [63]. They reported that treated ovarian cancer patients are positive for intrachromosomal rearrangements involving the P-GP gene, placing it under the control of the promoter of strongly transcript genes, with SLC25A40 being the most consistently involved. Therefore, stratification of patients according to P-GP fusion presence could lead to treatment with drugs that are not a substrate or are poor substrates of P-GP.

## 4. Molecular Tools to Assess HRD

At present, two available testing tools are being used for HRD assessment, Myriad MyChoice and FoundationOne CDX. The first is an HRD companion diagnostic test that generates a genomic instability score (GIS) through an NGS assay assessing loss of heterozygosis (LOH), telomeric allelic imbalance (TAI), and large-scale transitions (LST) (Myriad Genetic Laboratories; BRACAnalysis^®^ Technical Specifications 2012; available online at https://www.myriad.com/lib/technical-specifications/BRACAnalysis-Technical-Specifications.pdf (Myriad Genetic Laboratories, Updated: April 2012). In contrast, FoundationOne CDX not only generates an instability score predictive of HRD but can also detect mutations in a multi-gene panel of 324 genes, tumor mutational burden (TMB), and micro-satellites instability (MSI). The assay generates a LOH score calculated as the percentage of LOH in the tumor genome; a LOH score ≥ 14% or 16% is regarded as “LOH high” and indicates an HRD status [64]. Both assays have been used within several randomized clinical trials (RCTs) to assess HRD status in patients with HGSOC undergoing maintenance therapy with PARP-is both in first-line and relapse settings. The results of these RCTs show that HRD assays somehow fail to clearly distinguish those patients who benefit from PARP-is from those who do not. Some limitations of current validated HRD assays can be addressed to explain these findings. Firstly, the clinical context and timing of tumor HRD testing must be considered. Both Myriad MyChoice and FoundationOne CDX rely on genomic scars, intended as gross genomic rearrangements and aberrations representing a permanent fingerprint of HRD-related genomic instability. These molecular signatures are static by definition, and their assessment via GIS or LOH score relies on aberrations accumulated over time, irrespective of tumor evolution and current HRD status. Since surgery and biopsies of recurrent tumors are rarely performed at relapse, HRD testing is almost always performed on samples collected during primary upfront or interval debulking surgery. Therefore, The GIS or LOH score obtained reflects the chemo-naïve state of the tumor. Recent translational studies have shown spatial and temporal tumor heterogeneity in HGSOC. Subclonal tumor cell proliferation leads to the accumulation of different passenger gene mutations and to the acquisition of a mutational landscape that differs within an individual tumor and its metastasis. Moreover, platinum-based chemotherapy eradicates chemosensitive tumor cells and meanwhile selects those subclones resistant to it, leading to the development of temporal diversification of the recurrent tumor mutational landscape from the original one. As HRD cells are sensitive to platinum salts, chemotherapy leads to the selection of HRP clones; the further the patient is treated with platinum and the further the patient is from diagnosis, the higher the risk of discordance between actual HRR status and GIS/LOH score performed before first-line treatment. It is also noteworthy that HRD testing is usually performed on tumor biopsies, which represent only a fraction of the entire tumor proliferation; thus, a sample could be found to be HRD, while the tumor consists of a mix of HRD and HRP components, with the HRP component contributing to a reduced sensitivity or primary resistance to PARP-is. Secondly, HRD assessment with both Myriad MyChoice and FoundationOne CDX rely on a predefined cutoff, leading to difficulty in the interpretation of results next to the threshold. Myriad GIS cutoff of ≥42 was proposed in a study to detect tumors with BRCA1/2 mutations or BRCA1 promoter methylation [65]. However, Takaya et al. showed that if analyzed separately, BRCA1-2-mutated breast and ovarian cancer display a different distribution of HRD scores [66]. In line with Marquard and colleagues’ results [67], breast cancer samples show lower HRD scores compared to ovarian cancer, whereas BRCA-mutated ovarian cancer and especially HGSOC are more likely to have HRD scores > 63. Furthermore, the survival analysis of 537 cases of ovarian cancer revealed a better survival rate in patients with GIS > 63 and an overlap of survival curves in patients with HRD scores of 42–63 and <42, suggesting that a GIS > 63 should be used as the optimal threshold for HRD definition in HGSOC.

The use of an unfit threshold leads to the risk of misclassification; that is, the possibility to have either false positive or false negative results. The use of a higher cutoff would lead to a higher specificity of GIS increasing the false negatives rate. Conversely, the use of a lower threshold would minimize the risk of not treating with PARP-is patients who could benefit from them, but would also label as HRD-positive patients with an HRR-proficient tumor that would appear to be resistant to PARP-is. Recently, the European Network for Gynaecological Oncological Trials (ENGOT) promoted the European HRD ENGOT Initiative for the evaluation of various academic HRD tests designed by seven academic research laboratories [68]. The performance of each academic HRD test was assessed on tumor samples from the PAOLA-1 study, discriminating patients who benefited from PARP-is from those who did not, to provide an accurate and cost-effective test for HRD assessment. Preliminary data from two academic HRD tests were recently presented. Both the Geneva HRD test [69], developed by Geneva University Hospital in Switzerland and based on a CNV analysis using OncoScan™ CNV Assay, and NOGGO-GIS Assay [70], developed by Hamburg University in Germany and based on a hybrid capture NGS test, were validated with similar performance characteristics to the clinical trial assay, but with a significantly lower failure rate at a lower cost. They could be easily implemented in routine clinical practice, although further data are needed to evaluate if their application in other patient cohorts can overcome the limits met with the commercially currently available tests.

## 5. Strategies to Bypass Resistance

Several strategies are being developed to prevent or delay resistance to PARP-is. The most interesting are combinations with immune-checkpoint inhibitors (ICIs) and with targeted agents. Here, we describe the available data and the most promising future developments.

### 5.1. PARP Inhibitors and Chemotherapy

The combination of PARP-is and chemotherapeutic agents represents a potential strategy to overcome PARP-is resistance. A number of studies evaluating this combination were conducted (Table 2), including phase II and III trials [71,72,73,74,75]. Olaparib was the first PARP-i tested in combination with chemotherapy, but velaparib is the PARP-i mostly evaluated in combination with chemotherapy. Indeed, it is characterized by a lower PARP trapping activity and a better safety profile. Both PARP-is showed statistically significant improvements in PFS when combined with chemotherapy, especially in BRCA-mutated and HRD-positive HGSOC. The best chemotherapy combination seems to be carboplatin-paclitaxel. Conversely, disappointing results were observed with other agents, such as the topoisomerase-I inhibitor topotecan and the alkylating agent cyclophosphamide. The most frequent grade 3 or more adverse events were hematological. Several studies exploring these combinations are ongoing, both with olaparib (NCT03161132) and with veliparib (NCT01113957, NCT02483104, NCT01145430, NCT01459380, NCT00989651). This strategy is more a challenge to prevent the onset of primary resistance than to restore PARP-is sensitivity. Nevertheless, the safety profile represents an Achilles heel and requires careful consideration.

### 5.2. PARP Inhibitors and Immune Checkpoint Inhibitors

The rationale for the combination of ICIs and PARP-is is based on the evidence that HRD cancers have a higher tumor mutational burden (TMB), leading to elevated neo-antigen loads and causing an increased anti-tumor immune response [76]. Moreover, PARP-is can upregulate PD-L1 expression, thus enhancing the antitumor effect of the combination [77]. A number of clinical trials evaluated these combinations with mixed outcome results (Table 3).

Phase II trials studied both niraparib and olaparib in association with either pembrolizumab or durvalumab in platinum-sensitive relapsed (PSR) and platinum-resistant relapsed (PRR) ovarian cancer. Although the double combinations showed poor activity in PRR ovarian cancer [78,79,80,81], more interesting results were observed in PSR ovarian cancer [82,83,84]. More compelling data are expected from triplet combinations of PARP-is, ICIs and anti-angiogenetic, especially in BRCAwt and non-HRD tumors. Indeed, the phase II MEDIOLA study showed promising clinical activity of the combination of durvalumab, bevacizumab, and oaparib in gBRCA-negative PSR ovarian cancer. However, neither genomic instability nor PD-L1 was identified as predictive of survival [85]. Data from the first phase III study have been recently presented. The DUO-O trial is the first phase III trial to demonstrate clinical benefit with the addition of an ICI to a PARP-i in patients with newly diagnosed ovarian cancer. It investigated the combination of carboplatin and paclitaxel with bevacizumab and durvalumab followed by maintenance with bevacizumab, durvalumab, and olaparib in patients without sBRCA mutation. The preplanned interim PFS analysis showed a significant benefit with the triplet combination in the intention-to-treat (ITT) population (HR 0.63 [0.52–0.76], *p* < 0.0001), HRD-positive (HR 0.49 [0.34–0.69], *p* < 0.0001) and especially in the HRD-negative population (HR 0.68 [0.54–0.86]; 20.9 vs. 17.4 vs. 15.4 months in the subgroup treated with the triplet combination vs. bevacizumab alone vs. bevacizumab plus durvalumab, respectively). Safety was consistent with the profile of each agent, and no new adverse events were observed. Nevertheless, toxicity is not negligible. Indeed, 35% of the patients in the triplet arm experienced adverse events causing discontinuation of one or more of the triplet drugs [86]. A number of phase III clinical trials are ongoing to evaluate different PARP-is in combination with various ICIs, eventually in association with bevacizumab. They will probably help us to better manage toxicity and to understand which subgroups of patients could benefit most from these combinations (Table 4).

**Table 3 ijms-24-11890-t003:** PARP-is and ICIs. Clinical studies evaluating the combination of PARP-is and immune checkpoint inhibitors (ICIs); PARPi, PARP inhibitor; PSR, platinum-sensitive relapsed; PRR, platinum-resistant relapsed; O, olaparib; D, durvalumab; B, bevacizumab; N, niraparib; P, pembrolizumab; ORR, objective response rate; PFS, progression-free survival; OS, overall survival; AEs, adverse events; NA, not available).

Trial Name	Phase	PARPi	Number of Patients	Population	Treatment Arms	Primary Endpoint	ORR (%)	PFS Months	OS Months	Most Common G3-4 AEs (%)
MEDIOLA (NCT02734004) [82,83,85]	I/II	olaparib	32 + 31	BRCA wild type PSR ovarian cancer	O + D (doublet cohort); O + D + B (triplet cohort)	ORR; safety	31.3 vs. 77.4	5.5 vs. 14.7	NA	anemia (17.6), increased lipase (11.8), neutropenia (8.8), lymphopenia (8.8)
GINECO BOLD (NCT04015739) [78]	II	olaparib	74	PSR and PRR ovarian cancer	O + D + B	DCR	70 in PRR; 40 in PSR	4.1 in PRR; 4.9 in PSR	18.8 in PRR; 18.5 in PSR	NA
TOPACIO/KEYNOTE-162 (NCT02657889) [79]	I/II	niraparib	62	recurrent ovarian carcinoma, irrespective of BRCA mutation status	N + P	ORR	18	NA	NA	anemia (21); thrombocytopenia (9)
OPEB-01 (NCT04361370) [84]	II	olaparib	22	PSR BRCA Wild Type Ovarian Cancer	O + P + B	PFS rate of 6 months	68.2	NA	NA	NA
OPAL (NCT03574779) [80]	II	niraparib	41	PRR ovarian cancer	N + D + B	ORR	17.9	7.6	NA	hypertension (22), fatigue (17.1), and anemia (17.1)
MOONSTONE/GOG-3032 (NCT03955471) [81]	II	niraparib	41	PRR ovarian cancer	N + D	ORR	12	2.1	NA	NA

**Table 4 ijms-24-11890-t004:** Ongoing trials with PARP-is and ICIs. Phase III clinical trials evaluate different PARP-is in combination with various ICIs, some of these in association with bevacizumab (PARPi, PARP inhibitor; O, olaparib; D, durvalumab; B, bevacizumab; N, niraparib; R, rucaparib; Nivo, nivolumab; P, pembrolizumab; Dosta, dostarlimab; A, atezolizumab; CT, chemotherapy; PFS, progression-free survival; OS, overall survival).

Trial Name	Phase	PARPi	Number of Patients	Population	Treatment Arms	Primary Endpoint
ATHENA-COMBO (NCT03522246) [87]	III	rucaparib	1000	Newly diagnosed advanced ovarian cancer after first line CT	R + Nivo/placebo	PFS
DUO-O (NCT03737643) [86]	III	olaparib	1374	Newly diagnosed advanced ovarian cancer after first line CT	B + O/placebo + D/placebo	PFS in non-tBRCA HRD positive patients; PFS in all non-tBRCA patients
FIRST (NCT03602859) [88]	III	niraparib	1405	Newly diagnosed advanced ovarian cancer after first line CT	N/placebo + Dosta/placebo	PFS for PD-L1 positive participants; PFS for all participants
KEYLYNK-001 (NCT03740165) [89]	III	olaparib	1367	Newly diagnosed advanced BRCA wild type ovarian cancer	CT + P/placebo followed by P/placebo + O/placebo	PFS for PD-L1 positive participants; PFS for all participants
ANITA (NCT03598270) [90]	III	niraparib	414	PSR ovarian cancer	Platinum-base CT + A/placebo followed by N + A/placebo	PFS
NItCHE-MITO33 (NCT04679064) [91]	III	niraparib	427	Recurrent ovarian cancer not candidate for platinum-base CT	CT vs. N + Dosta	OS

### 5.3. PARP Inhibitors + Anti-Angiogenic Agents

Angiogenesis plays a key role in ovarian cancer pathogenesis. Preclinical data have evidenced synergy between PARP-is and anti-angiogenic agents. Two angiogenesis inhibitors with different mechanisms of action, bevacizumab, and cediranib, have demonstrated antitumor activity in patients with advanced ovarian cancer [92,93]. Bevacizumab is a humanized anti-VEGF monoclonal antibody approved in association with chemotherapy and as maintenance for both newly diagnosed and recurrent ovarian cancer, since it showed PFS improvement compared to chemotherapy alone [94,95]. It has been evaluated in association with PARP-is in both phase II and III trials. The AVANOVA2 trial, a randomized phase II study, evaluated niraparib plus bevacizumab vs. niraparib alone for PSR ovarian cancer. PFS was improved with niraparib plus bevacizumab compared to niraparib alone in the subgroup of patients with gBRCA wild-type (11.3 vs. 4.2 months, HR = 0.32), and a positive trend was observed for BRCA-mutated patients (14.4 vs. 9.0 months, HR = 0.49). PAOLA-1 is the first randomized, double-blind, phase III trial that evaluated the combination of a PARP-i and an antiangiogenic agent as maintenance treatment in the first-line setting. The study enrolled patients with advanced, high-grade ovarian cancer in response following first-line chemotherapy with carboplatin and paclitaxel plus bevacizumab, regardless of BRCA mutation status. The study randomized patients to olaparib plus bevacizumab or placebo plus bevacizumab. The PFS was significantly longer in the combination arm compared to the bevacizumab plus placebo arm (HR 0.59, 95% CI 0.49–0.72; *p* < 0.001). The advantage was especially pronounced in the HRD-positive tumors, since median PFS was 37.2 vs. 17.7 months (HR 0.33, 95% CI 0.25–0.45), while median PFS was similar in the HRD-negative tumors (16.6 vs. 16.2 months for the olaparib and the placebo arm, respectively) [96]. At 5 years, updated PFS was still significantly longer in the experimental arm compared to the placebo arm in patients with HRD-positive tumors (46.8 vs. 17.6 months; HR 0.41, 95% CI 0.32–0.54). Furthermore, albeit the difference in OS was not statistically significant in the ITT population, the median OS was significantly prolonged with the olaparib and bevacizumab combination in the HRD-positive group (median 75.2 vs. 57.3 months; HR 0.62; 95% CI 0.45–0.85) [97]. These interesting results led to FDA and EMA approval of olaparib plus bevacizumab as maintenance therapy for patients with HRD-positive tumors.

A similar study in the same setting, MITO 25 (NCT03462212), is ongoing. It is a randomized, three-arm, molecular-driven phase II trial in patients with advanced ovarian cancer. According to HRD status, patients are randomized to receive first-line carboplatin plus paclitaxel in association with bevacizumab followed by bevacizumab maintenance vs. the same chemotherapy regimen, followed by rucaparib vs. chemotherapy plus bevacizumab followed by bevacizumab plus rucaparib. Another promising antiangiogenetic agent is cediranib, an oral receptor tyrosine kinase inhibitor of all three VEGF receptors (VEGFR-1, 2 and 3). It has been evaluated in association with olaparib in both phase II and III trials with contradictory results. Indeed, a randomized phase II study (NCT01116648) evaluated the combination of cediranib and olaparib vs. olaparib alone in PSR ovarian cancer and showed a significant PFS increase (17.7 vs. 9.0 months, HR 0.42, *p* = 0.005). It is noteworthy that the most significant benefit in PFS occurred in gBRCA wild-type patients (16.5 vs. 5.7 months, *p* = 0.008) [98]. Unfortunately, a confirmatory phase III trial (NRG-GY004) failed to confirm a PFS improvement. However, the PFS HR was 0.55 favoring olaparib plus cediranib and 0.63 favoring olaparib alone compared to chemotherapy in patients with gBRCA mutant tumors [99]. A few phase II studies tested the combination of cediranib and olaparib in PRR ovarian cancer patients, either in single arm (CONCERTO, EVOLVE) or compared with chemotherapy (BAROCCO, OCTOVA, AMBITION). Although ORR and PFS improvement are modest in the overall population, the most promising results were reached with a biomarker-driven approach. In fact, in the AMBITION trial, the olaparib cediranib combination demonstrated a 50% ORR in the HRD-positive tumors subgroup [100,101,102,103,104].

Notwithstanding the lack of superiority of the combination over the single agent in terms of OS and the contradictory results in the overall population, this association could be a viable alternative for patients not suitable for or opposed to chemotherapy. Furthermore, it could be a feasible option in PRR ovarian cancer patients and in patients resistant to PARP-is according to the mechanism of resistance. Further phase III studies are ongoing evaluating the cediranib–olaparib combination, both as maintenance following platinum-based chemotherapy in PSR (ICON9, NCT03278717) and as a treatment arm compared to non-platinum-based chemotherapy in PRR ovarian cancer (NRG-GY005, NCT02502266). Another field of growing interest is the combination of PARP-is, antiangiogenetic therapy and ICIs, as mentioned above.

### 5.4. PARP Inhibitors + BET Inhibitors

The bromodomain and extra-terminal domain (BET) proteins have a crucial role in epigenetic gene regulation and induce oncogene transcription. BET inhibitors cause indirect inhibition of HR, reduce transcription of BRCA1 and RAD51, and cause re-sensitization to PARP-is. Moreover, BET and PARP inhibitors showed synergistic activity, as BET inhibitors enhance PARP-induced DNA damage. Therefore, this combination could become a therapeutic strategy also in HR-proficient tumors [105,106]. Based on these premises, early phase trials are underway to investigate this association, both as phase I (NCT03205176) and phase II trials (NCT05071937, NCT05327010, NCT05252390) in patients with advanced solid tumors, including ovarian cancer.

### 5.5. PARP Inhibitors + Radiation Therapy

Preclinical studies showed that PARP-is sensitize cancer cells to ionizing radiation. Indeed, irradiation of cancer cells during treatment with PARP-is increases DSB [107,108]. Furthermore, experience from other cancers, such as lung cancer, supports the efficacy of local treatment for oligo-progressive disease maintaining the same systemic therapy. Therefore, combining PARP-is and radiotherapy represents a promising strategy to enhance DNA damage and prolong PARP-is’ benefit. However, feasibility depends on the size and number of metastases. Thus, the treatment approach must be personalized and discussed by a multidisciplinary team. Moreover, safety remains a challenge, even if more modern radiation techniques, such as proton therapy and intensity-modulated radiotherapy, promise to be more accurate and to spare healthy tissues. Two phase I trials are ongoing to determine the maximum tolerated dose (MTD) and safety of talazoparib (NCT03968406) and veliparib (NCT01264432) in combination with radiation in gynecologic tumors, including ovarian cancer.

## 6. ATR Inhibitors

ATR is a kinase that responds to many types of stress. It controls cell cycle checkpoint activation and ensures cell cycle arrest in response to DNA damage [109,110]. Thus, ATR represents a good target as its inhibition promotes DNA damage harboring cells entering premature mitosis. ATR inhibitors are being investigated as monotherapy and in combination. The most studied ATR inhibitor is ceralasertib, which was evaluated in combination with olaparib in a phase 2 trial in patients with PRR EOC with scarce activity [111]. Ceralasertib was also studied in patients with PSR ovarian cancer who gained benefit from a PARP-i (the DUETTE study). Unfortunately, the trial was withdrawn due to the lack of efficacy of the ceralasertib–olaparib combination in patients with triple-negative breast cancer (NCT03330847). Interestingly, ceralasertib showed a consistent antitumor activity in combination with chemotherapy in cell lines [112]. Consequently, a modular phase I/1b study is ongoing evaluating increasing doses of ceralasertib in combination with carboplatin, olaparib, or durvalumab in patients with advanced malignancies, including BRCA-mutant, RAD51C/D-mutant or HRD-positive status PSR ovarian cancer patients who have previously progressed on a licensed PARP-i (NCT02264678). Besides ceralasertib, there are a number of ATR inhibitors under clinical development. They are being investigated in phase 1 and 2 trials. The most promising agents are berzosertib, elimusertib, and camonsertib.

## 7. CHK1 Inhibitors

Downstream of ATR, CHK1 is involved in the cell cycle arrest, thereby causing fork collapse in S-phase cells and cell death. Combinations of ATR and Chk1 inhibitors with olaparib were evaluated in both HR-deficient and HR-proficient murine ovarian cells resistant to olaparib. They were synergistic in sensitive and resistant sublines, regardless of HR status [113,114]. In a phase 1 study, the CHK1 inhibitor prexasertib in combination with olaparib showed preliminary clinical activity achieving 4 partial responses out of 18 patients with BRCA-mutated ovarian cancer [115]. Moreover, the combination of the Chk1 inhibitor LY2880070 with low-dose gemcitabine showed antitumor activity in patients with advanced or metastatic HGSOC with a disease control rate of 59.3% and a manageable safety profile [116].

## 8. WEE1 Inhibitors

WEE1 is a kinase that regulates G2/M progression, interrupts the cell cycle, and provides time for DNA damage repair. Inhibition of WEE1 can re-sensitize BRCA-deficient cells to PARP-is. WEE1 inhibitors have been evaluated in combination with a PARP-i as in the EEFORT trial, a phase II, noncomparative study of oral adavosertib with or without olaparib in women with PARP-resistant ovarian cancer. The combination of adavosertib and olaparib showed an ORR of 29%, while adavosertib alone had an ORR of 23%. The median PFS was 6.4 months and 5.5 months, respectively. Similarly, patients achieved an 89% and a 63% clinical benefit rate (CBR) in the combination arm and in the adavosertib alone arm, respectively. The clinical benefit was irrespective of the BRCA status [117]. The trial is still ongoing and recruiting, and additional translational analysis is ongoing to identify potential predictive factors (NCT03579316).

Furthermore, adavosertib was evaluated in combination with chemotherapy agents to exploit the high sensitivity of HGSOC to chemotherapy and to increase the mitotic effect of the WEE1 inhibitor. The first randomized trial was a phase 2, double-blind, placebo-controlled, multicentre trial that evaluated gemcitabine plus adevosertib or placebo in patients with PRR ovarian cancer. The primary endpoint was met, as the combination showed a statistically significant improvement of PFS (median 4.6 vs. 3 months, hazard ratio 0.55 [95% CI 0.35–0.90]; log-rank *p* = 0.015). Moreover, median OS was significantly longer with adavosertib plus gemcitabine (11.4 months) than with placebo plus gemcitabine (7.2 months; HR 0.56 [95% CI 0.35–0.91]; log-rank *p* = 0.017). The most common grade ≥ 3 adverse events were hematological, and the incidence was significantly higher in the combination arm [118]. Because of these intriguing results, adavosertib was evaluated in combination with other chemotherapeutic agents in a phase 2 trial with promising results [119]. Therefore, it is appealing to combine WEE1 or ATR inhibitors with PARP-is or chemotherapy. Nonetheless, toxicity is a concern. Next-generation WEE1, ATR or PARP-is with improved specificity could help solve this challenge. Additionally, sequential rather than concurrent administration could be an alternative efficacious schedule.

## 9. Indirect Inhibition of HR

Another potential strategy to overcome resistance to PARP-is is indirectly inhibiting HR. This can be achieved with agents targeting the PI3K-AKT pathway that has been proved to impair HR. These targeted therapies have been developed and reported to have synergistic activity with PARP-is.

Two phase 1 studies evaluated the combination of olaparib and the AKT inhibitor capivasertib, and both recommended the doses of olaparib 300 mg twice a day with capivasertib 400 mg twice a day 4 days on, 3 days off. A phase 1 trial evaluated the combination in 64 patients with advanced solid tumors, most commonly ovarian cancer. The combination demonstrated a CBR of 44.6%. Eleven (44%) of 25 patients with ovarian cancer and 7 of 10 patients with gBRCA1/2-mutant ovarian cancer achieved clinical benefit [120]. In the second phase 1 study, 38 patients were enrolled, including 16 (42%) of patients with ovarian cancer, of which 87% were platinum-resistant or refractory. Of 32 evaluable patients, 19% achieved ORR. Six (43%) of 16 ovarian cancer patients had clinical benefits [121]. The dose expansion phases of both studies are currently ongoing.

Although PI3K inhibitors alone showed poor activity in ovarian cancer, promising results derive from combination strategies, mostly in PRR ovarian cancer. A phase III trial is ongoing evaluating the efficacy of the olaparib–alpelisib combination in platinum-resistant or refractory HGSOC, with no germline BRCA mutation detected (NCT04729387).

Likewise, the inhibition of heat shock protein 90 (HSP90) induces HR deficiency and results in restored sensitivity to PARP-is. The HSP90 inhibitor ganetespib has been evaluated in a multicenter phase I/II trial in association with weekly paclitaxel in platinum-resistant ovarian cancer. Two (20%) of 10 patients achieved a partial response, and 4 (40%) patients had a stable disease [122]. The Eudario trial is a phase II multicenter clinical trial to evaluate ganetespib in association with niraparib in platinum-sensitive ovarian cancer. The study is no longer recruiting, and results are awaited (NCT03783949).

## 10. RAS/RAF/MEK

Another indirect inhibition of HR involves the RAS/RAF/MEK pathway. The SOLAR trial is a phase 1/2 trial evaluating the combination of the MEK inhibitor selumetinib plus olaparib in advanced or recurrent solid tumors (NCT03162627).

## 11. POLQ = DNA Polymerase θ

A research field of increasing interest consists of inhibitors of the DNA polymerase θ (POLQ). HR-deficient cells depend on the microhomology-mediated end-joining (MMEJ) pathway for DNA DSB repair, and this pathway is driven by POLQ. Therefore, POLQ inhibitors may be effective in both tumors resistant to PARP-is and tumors naïve to PARP to prevent or delay the onset [123,124]. Novobiocin (NVB) is an antibiotic that inhibits the ATPase activity of POLQ. NVB selectively kills HR-deficient cells in vitro and in vivo. In a patient-derived xenograft (PDX) model, NVB in association with olaparib demonstrated to induce complete tumor regression. Finally, NVB reduces tumor growth in a PDX model with BRCA1 and 53BP1 loss of function [125]. Clinical trials are necessary to validate the role of POLQ inhibitors, to explore the safety profile and establish the best setting to prevent the onset of resistance, either in PARP-is-resistant or in PARP-is naïve tumors.

## 12. Rechallenge

Due to the efficacy of PARP-is after response to platinum-based chemotherapy, retreatment with PARP-is has captured much interest. Rechallenge with olaparib has shown statistically significant benefits in BRCA-positive EOC patients in the prospective OREO trial. However, an absolute PFS improvement of only 1.5 months was not considered clinically relevant by the international community [126]. Likewise, a small retrospective study evidenced a scarce benefit with retreatment. The median PFS was 8.5 months in patients who obtained a complete remission after chemotherapy, and 5.5 months in patients with partial remission [127]. Although the rechallenge with PARP-is cannot be considered a standard in all ovarian cancer patients, it would be useful to identify which patients could benefit from re-treatment.

## 13. Conclusions

Despite the efficacy of PARP-is in the treatment of ovarian cancer, overcoming resistance currently represents a big challenge. To date, combinations of PARP-is and either anti-angiogenetics or ICIs have achieved the most promising results. Nevertheless, toxicity is not negligible. New drugs targeting different pathways, including the ATR-CHK1-WEE1, the PI3K-AKT, and the RAS/RAF/MEK, are under development, and the results are eagerly awaited. The deep knowledge of the molecular basis of PARP resistance requires the use of advanced technologies to test new molecules and improve clinical outcomes by reducing toxicity.

## Figures and Tables

**Figure 1 ijms-24-11890-f001:**
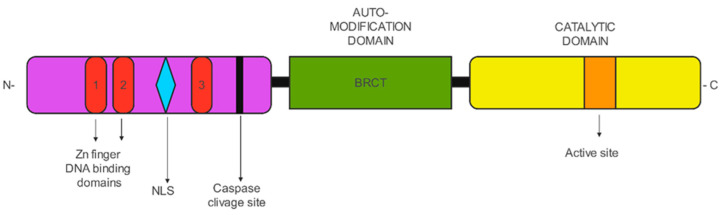
PARP1 domains. DNA-binding and catalytic domains. Three zinc finger (ZnF) domains at N-terminus followed by the BRCT (breast cancer-associated C-terminal) domain, WGR (Trp-Gly-Arg rich) domain in the central region, and helical subdomain (HD) and the signature ADP-ribosyl transferase (ART) subdomain, together called the catalytic (CAT) domain at the C-terminus.

**Figure 2 ijms-24-11890-f002:**
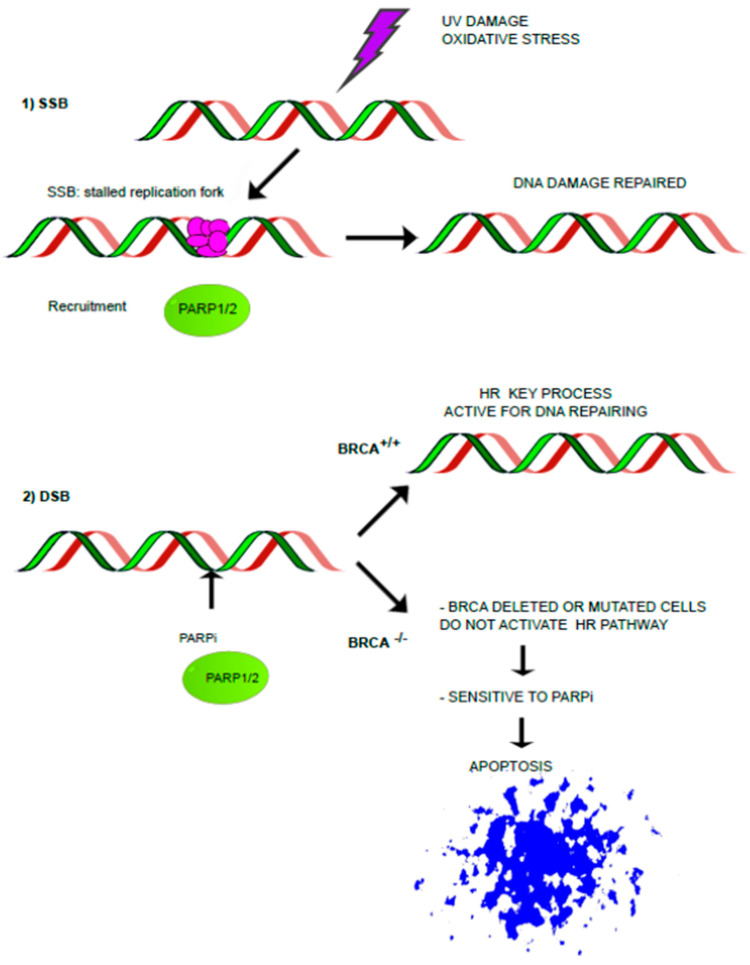
DNA damage repairing mechanism. In proliferating cells, single-stranded breaks (SSBs) are frequently encountered and SSBs are primarily repaired by the PARP-dependent basic excision repair (BER) pathway “(**1**)”. A double-strand break (DSB) can occur as a consequence of unresolved replicative stress. To repair it, homologous recombination (HR) is activated where the main effectors are BRCA1/2 restoring genomic integrity. If mutated BRCA1/2 is present, HR does not occur. Therefore, HR-deficient cancer cells in association with PARP-is will move towards apoptosis “(**2**)”.

**Table 1 ijms-24-11890-t001:** Reversion mutations of BRCA1. The mutations listed below were detected by Lin et al. in the cfDNA of one patient [39].

	The Nucleotide and the Protein Sequence Changes	Allele Frequencies
Reversion mutation #1	c.1046A > G (p.E349G)	7.7%
Reversion mutation #2	c.1047G > T (p.E349D)	0.45%
Reversion mutation #3	c.1039_1077del39 (p.L347_P359del)	0.16%
Reversion mutation #4	c.1035_1055del21 (p.D345_K351del)	0.13%

**Table 2 ijms-24-11890-t002:** PARP-is and chemotherapy. Clinical trials evaluating the combination of PARP-is and chemotherapeutic agents that represents a potential strategy to overcome PARP-is resistance (HGSOC, high-grade serous ovarian cancer; PARPi, PARP inhibitor; ORR, objective response rate; PFS, progression-free survival; OS, overall survival; AEs, adverse events; ITT, intention-to-treat; NA, not available).

ClinicalTrials.gov Identifier	Phase	PARPi	Number of Patients	Population	Treatment Arms	Primary Endpoint	PFS Months	OS Months	Most Common G3-4 AEs (%)
NCT01081951 [71]	II	olaparib	162	Platinum-sensitive, recurrent HGSOC, with or without BRCA 1/2 mutations	Olaparib plus CT → olaparib maintenance vs. CT	PFS	12.2 vs. 9.6 (HR 0.51 [95% CI 0.34–0.77]; *p* = 0.0012)	33.8 vs. 37.6 months (HR 1.17 [95% CI 0.79–1.73]; *p* = 0.44)	Neutropenia (43 vs. 35); anaemia (9 vs. 7)
NCT0247058 [72]	III	veliparib	1140	Untreated HGSOC	CT + placebo →placebo vs. CT + veliparib →placebo vs. CT + veliparib → veliparib	PFS in veliparib-throught group vs. control group	23.5 vs. 17.3 in ITT (HR 0.68; 95% CI, 0.56 to 0.83; *p* < 0.001)	NA	Neutropenia (58 vs. 49); anemia (38 vs. 26); thrombocytopenia (28 vs. 8)
NCT01690598 [74]	I/II	veliparib	27	relapsed advanced platinum resistant or partially platinum sensitive HGSOC with negative or unknown BRCA1/2 status	Veliparib plus topotecan	ORR	2.8 (95% CI [2.6–3.6])	7.1 months (95% CI [4.8–10.8])	Infection (22.2); neutropenia (11.1); fatigue (7.4); nausea (7.4)
NCT01306032 [75]	II	veliparib	75	Pretreated BRCA-mutant ovarian cancer	Veliparib plus cyclophsphamide vs. cyclophsphamide	ORR	2.3 vs. 2.1 (*p* = 0.68)	NA	Lymphopenia (17.3 vs. 4); anemia (2.7 vs. 0); neutropenia (2.7 vs. 2)

## Data Availability

Not applicable.

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
