# Peer review of "An Overview of PARP Resistance in Ovarian Cancer from a Molecular and Clinical Perspective"

_ijms, 2023, doi:10.3390/ijms241511890_

Round 1

Reviewer 1 Report

Thank you for the opportunity to evaluate such an interesting and comprehensive review article. In my opinion, such article should be published in IJMS due to the importance of the topic. 

Minor editing of English language required.

Author Response

We thank the reviewer for her/his comments. We carefully re-read the paper and edited the English language.

Reviewer 2 Report

The authors of the review have extensively reviewed literature on epithelial ovarian cancer (EOC), particularly its primary subtype, high-grade serous carcinoma (HGSOC). Furthermore, the authors have delved into the molecular characterization of HGSOC, uncovering a remarkable finding. Authors also acknowledged the prevalent issue of drug resistance in EOC treatment. They have highlighted the ongoing investigation of various strategies aimed at overcoming this challenge. 

In addition to exploring the therapeutic potential of PARP-is, the authors have discussed the development of novel drugs that target different pathways implicated in EOC. These include the ATR-CHK1-WEE1, PI3K-AKT, and RAS/RAF/MEK pathways.

Overall, the authors' comprehensive study provides valuable insights into EOC, shedding light on its treatment options, the role of molecular characterization, the challenge of drug resistance, and the potential avenues for future research and innovation.

Here are few minor comments:

1. the quality of Table 2 can be improved, texts are not clearly visible.

2. The influence of reversion mutations on PARPi resistance needs to be explained and the list of the mutations should be mentioned if available.

3. Figure and Table legends can be elaborated

The English used in this review paper is clear, concise, and well-structured. The sentences are coherent and effectively convey the key points discussed in the paper. The language used is technical but still accessible, allowing the reader to understand the subject matter without overwhelming them with complex terminology.

Author Response

Authors response:

We thank reviewer 2 for the compliments on this review.  We have really appreciated her/his useful suggestions

  • We have added a Word file containing Table 2 so that the journal can lay it out in the best possible way without losing the resolution.
  • We explained the role of reversion mutations on PARPi resistance as suggested. Moreover, we reported the mutations found by Lin et al in one patient with platinum-resistant cancer detected in cfDNA.
  • The captures of the Figures and Tables have been elaborated and are in the main text in track- changes.
